

# Recruitment of cognitive control regions during effortful self-control is associated with altered brain activity in control and reward systems in dieters during subsequent exposure to food commercials

Richard B. Lopez[1,*], Andrea L. Courtney[2,*] and Dylan D. Wagner[3]

[1] Department of Psychological Sciences, Rice University, Houston, TX, United States of America
[2] Department of Psychology, Stanford University, Stanford, CA, United States of America
[3] Department of Psychology, The Ohio State University, Columbus, OH, United States of America
[*] These authors contributed equally to this work.

Corresponding author
Richard B. Lopez,
richard.lopez@rice.edu

## ABSTRACT

Engaging in effortful self-control can sometimes impair people's ability to resist subsequent temptations. Existing research has shown that when chronic dieters' self-regulatory capacity is challenged by prior exertion of effort, they demonstrate disinhibited eating and altered patterns of brain activity when exposed to food cues. However, the relationship between brain activity during self-control exertion and subsequent food cue exposure remains unclear. In the present study, we investigated whether individual differences in recruitment of cognitive control regions during a difficult response inhibition task are associated with a failure to regulate neural responses to rewarding food cues in a subsequent task in a cohort of 27 female dieters. During self-control exertion, participants recruited regions commonly associated with inhibitory control, including dorsolateral prefrontal cortex (DLPFC). Those dieters with higher DLPFC activity during the initial self-control task showed an altered balance of food cue elicited activity in regions associated with reward and self-control, namely: greater reward-related activity and less recruitment of the frontoparietal control network. These findings suggest that some dieters may be more susceptible to the effects of self-control exertion than others and, whether due to limited capacity or changes in motivation, these dieters subsequently fail to engage control regions that may otherwise modulate activity associated with craving and reward.

## INTRODUCTION

Broadly defined, self-regulation refers to the human capacity to flexibly regulate thoughts, emotions, and behaviors. Self-control, in particular, describes those situations in which self-regulatory processes are engaged to inhibit prepotent or automatic impulses, especially when these impulses conflict with other goals (*Carver, 2005*; *Heatherton & Wagner, 2011*;

*Hofmann, Friese & Strack, 2009*). In situations when impulses and cravings are relatively weak, less effort is required to engage in self-control. However, at other times, self-control can be experienced as effortful. A large body of research in experimental psychology has examined how the effects of exerting self-control in one domain can lead to self-regulation failure in other domains, resulting in the hypothesis that self-regulation is dependent on limited cognitive resources that can become depleted over subsequent self-regulation attempts (*Baumeister et al., 1998*; *Baumeister, Vohs & Tice, 2007*; *Muraven, Tice & Baumeister, 1998*; *Vohs & Heatherton, 2000*).

Most recently, the consensus appears to be that the available evidence, in favor of and/or against self-regulatory depletion effects, is inconclusive (*Friese et al., 2018*). However, a recent pre-registered study with large samples (Ns ≥ 377) has shown reliable effects consistent with an attentional-based account of depletion (*Garrison, Finley & Schmeichel, 2018*), and studies using more ecologically valid designs have provided supporting evidence for the role of self-regulatory fatigue in precipitating self-control failures in daily life (*Hofmann et al., 2012*; *Wilkowski et al., 2018*), suggesting, perhaps, that focusing on more ecologically valid and appetitive behaviors (such as dieting failures) may provide better traction on the underlying phenomena.

One population that may be particularly susceptible to self-control fatigue is chronic dieters (i.e., those who constantly monitor and attempt to control their food intake) as they frequently demonstrate weight fluctuations, weight gain, and self-control lapses (*Heatherton, Polivy & Herman, 1991*; *Lowe et al., 2006*; *Lowe et al., 2013*), including after effortful self-control exertion (*Friese, Engeler & Florack, 2015*; *Kahan, Polivy & Herman, 2003*; *Vohs & Heatherton, 2000*). Much of this prior work has generally treated dieters as a homogenous group, comparing these to non-dieters or to groups of dieters who were not "fatigued" by self-control exertion (e.g., *Wagner et al., 2013*). However, more recent work has taken an individual differences approach to better understand why *some* dieters may be more prone to self-control failure than others. For example, prior work in our lab has shown that those dieters who experience weaker food cravings and more positive mood in daily life are also most successful in controlling impulses to eat (*Lopez et al., 2016*). Another study revealed that after initial exertion of (effortful) self-control, dieters variably recruited brain regions associated with self-control and reward when exposed to appetizing food cues, and these differences predicted self-control outcomes in daily life (*Lopez et al., 2017*). However, this study did not measure brain activity during the initial self-control task. So, a key unanswered question is: are individual differences in recruitment of self-control brain regions during self-control exertion in dieters related to altered patterns of activity in brain regions associated with control and reward when dieters are subsequently exposed to appetizing food cues? Additionally, might a relationship between brain activity during self-control exertion and later exposure to tempting food cues itself predict self-regulation failures (i.e., overeating) in more ecologically valid settings?

To address these questions, we used functional neuroimaging in the present study to examine individual differences in the neural correlates of effortful self-control among chronic dieters and related activation in prefrontal regions associated with response inhibition to subsequent recruitment of control and reward-related brain areas during

exposure to appetitive food commercials. In addition to their constant deployment of self-control, the dieting population is characterized by a motivation to regulate food intake, allowing for cleaner operationalization of self-control success and failure. In a recent brain imaging study, dieters randomly assigned to first complete an effortful self-control task showed, on average, significantly higher activity in orbitofrontal cortex (OFC), a key region in the brain's reward system; (*O'Doherty, 2004*; *Suzuki, Cross & O'Doherty, 2017*; *Haber & Knutson, 2010*) during subsequent exposure to food cues, as well as reduced functional coupling between inferior frontal gyrus (IFG), an area of lateral prefrontal cortex, and the OFC—relative to dieters assigned to the control condition (*Wagner et al., 2013*). These findings indicated that when dieters engage in effortful self-control, the reward value of food may become amplified while self-control capacity—as indexed by reduced functional connectivity between OFC and IFG—may be compromised.

Work in cognitive neuroscience has identified lateral prefrontal cortex as a robust neural correlate of self-control, indexing one's capacity to engage in response inhibition (see *Aron, Robbins & Poldrack, 2014* for a review). Lateral prefrontal cortex has also been implicated in successful resistance of desires to smoke cigarettes as well as regulate food cravings in daily life (*Berkman, Falk & Lieberman, 2011*; *Lopez et al., 2014*). Additionally, several studies have found a general pattern such that, following initial exertion of self-control, activity in lateral prefrontal cortex subsequently decreases on subsequent tasks (*Friese et al., 2013*; *Hedgcock, Vohs & Rao, 2012*; *Luethi et al., 2016*; *Persson, Larsson & Reuter-Lorenz, 2013*) and the magnitude of this decrease has been found to be correlated with performance deficits in subsequent cognitive tasks (*Friese et al., 2013*; *Persson, Larsson & Reuter-Lorenz, 2013*). Moreover, these effects may be most pronounced in contexts where people are required to inhibit responses to stimuli with high reward value (*Freeman & Aron, 2016*).

Given these previous lines of work on self-control among dieters and the role prefrontal cortex plays in effortful self-control in the cognitive domain, we hypothesized that individual differences in dieters' recruitment of lateral prefrontal cortex during initial self-control exertion would be associated with: (1) altered brain responses during subsequent exposure to food commercials, and (2) ad libitum eating patterns following a later diet breaking episode. To test these hypotheses, we first measured brain activity as dieters performed an effortful self-control task in which they were required to actively inhibit reading a series of words that appeared on the screen over the course of seven minutes. Following this, participants engaged in a food-cue reactivity task involving food commercials that has previously been shown to reliably recruit the brain's reward system (*Rapuano et al., 2016*). This dual-task design allowed us to examine the correspondence between task-elicited activity in the lateral prefrontal cortex during self-control exertion in the first task and the balance between regions associated with self-control and those associated with food-reward in the second task. Here, we focus specifically on the relative balance of activity in regions associated with self-control and reward during exposure to food commercials, as our previous work has associated this balance measure with failure to resist the desire to eat in daily life, as measured by experience sampling (*Lopez et al., 2017*). As some have recently argued that self-control capacity may be better captured by the coordination of whole brain systems that support regulatory processes (e.g., frontoparietal

control network), rather than activity of discrete, independent regions (*Kelley, Wagner & Heatherton, 2015*), this balance measure was calculated using a systems-based approach. Following the procedure used previously in Lopez and colleagues (2017), we used independently defined, *a priori* regions/systems of interest, namely the frontoparietal control network, which enables flexible exertion of self-control (*Power et al., 2011*), and key regions of the reward system, namely OFC and ventral striatum, both of which reliably activate to appetizing food images (*Courtney et al., 2018*; *Demos, Heatherton & Kelley, 2012*; *Rapuano et al., 2015*; *Rapuano et al., 2016*; *Wagner et al., 2013*).

To constrain our hypotheses about potential relationships between PFC activity during effortful self-control and subsequent recruitment of control (vs. reward) regions during food cue exposure, we based our hypotheses on prior neuroimaging studies (*Luethi et al., 2016*; *Persson, Larsson & Reuter-Lorenz, 2013*) that suggest that following (effortful) exertion of self-control, there is less recruitment of lateral prefrontal cortex, compared to control groups that did not exert self-control, in a subsequent task requiring self-regulation. Thus, in line with this work we focused on individual differences among a group of dieters and hypothesized that dieters showing *more* recruitment of prefrontal cortex during effortful self-control, indexing task difficulty and/or the experience of effort, would subsequently show *less* recruitment of the frontoparietal control network and therefore *more* reward-related activity in OFC and VS when viewing food commercials—thus serving as evidence of a failure to appropriately engage self-control systems when confronted with appetitive food stimuli. Finally, we also hypothesized that higher PFC activity during effortful self-control, and/or lower frontoparietal (vs. reward) balance scores, would be associated with more disinhibited eating following a diet breaking episode—as measured in separate experimental session with the same participants.

## MATERIALS & METHODS

### Participants

Thirty-two female dieters ($M_{age} = 19.48$, $SD_{age} = 1.11$) were recruited from the Dartmouth community to participate in a two-part neuroimaging study for course credit. Dieting status was assessed by the Restrained Eating Scale (*Heatherton et al., 1988*; *Herman & Polivy, 1980*), and all participants were prescreened to ensure that they were actively dieting at the time the study was conducted. The study consisted of an initial fMRI scanning session in which brain activity was measured both during an effortful self-control task (*Wagner et al., 2013*) and a food cue reactivity task that used naturalistic food stimuli (i.e., fast food commercials; *Rapuano et al., 2015*). Approximately a week later, participants returned to the lab for a follow-up behavioral session, in which their diets were broken and subsequent disinhibited eating was measured, as per a previously validated diet-breaking procedure (*Demos, Kelley & Heatherton, 2011*; *Timko, Juarascio & Chowansky, 2012*; *Tomiyama et al., 2009*). All participants gave informed consent in accordance with guidelines set by the Committee for the Protection of Human Subjects at Dartmouth College, and were fully debriefed at the end of the study (IRB approval #20325).
## Procedure

In order to ensure participants remained naïve to the goals of the experiment, they were instructed that the study was primarily about attention and perception. Specifically, the experimenter told each participant the following: "We're interested in the relationship between attention and perception. Specifically, we will be exploring multiple types of perceptual processes, from higher-level person perception to lower-level sensory perception. Today we will be scanning your brain during an attention task, which will be followed by an episode of the popular TV show *Big Bang Theory*, as you would see it on TV with commercials. We are interested in how you perceive these popular TV characters. During the second visit, we'll have you do some tests of lower level perception involving different senses, such as vision, sight, smell, and taste."

Participants first completed an effortful self-control task adapted from Wagner and colleagues' (*2013*) study that was modified to be amenable to neuroimaging analysis (i.e., jittered presentation of stimuli and explicit baseline periods were added). For this task, all participants were told that they would watch a clip from a nature documentary featuring Canadian bighorn sheep, and at times various words would appear and move around on the screen, but they were told to "avoid reading these distractor words whenever they appear." On average, dieters find this task significantly more effortful, compared to a control condition in which they just watched the film and read the words as they pleased, Cohen's $d = 1.04$ (calculated from *Wagner et al., 2013*). The video lasted for seven minutes, with a jittered presentation of 40 words (all one-syllable and of neutral valence, e.g., "CHAIR" and "BOOK") throughout the task epoch.

Following the effortful self-control task, participants watched a complete episode of the popular television sitcom *The Big Bang Theory*. In between episode segments, participants viewed sets of commercials, with a total of 12 food commercials and 12 non-food (control) commercials that were matched for duration and level of engagement and interest (see Fig. 1 in *Rapuano et al., 2015* for a depiction of the design).

Food commercials featured menu items at popular fast food restaurants, such as McDonald's, Wendy's, and Kentucky Fried Chicken. In contrast, control commercials featured various products that were not directly consumptive or appetitive (e.g., 4G LTE cellular service, cars, cleaning products, shaving supplies). At the conclusion of the scanning session, we asked participants to report on their experiences of the effortful self-control control task. Specifically, they provided ratings on a 1–7 scale in response to questions that asked: "How difficult did you find this task?" and "How much did this task 'tire you out' or make you feel mentally exhausted/fatigued afterwards?"

Approximately one week after the scanning session, participants returned to the lab for a follow-up behavioral session, consisting of a commonly used milkshake preload manipulation (*Brace, Crombag & Yeomans, 2016*; designed to temporarily break participants diets; *Demos, Kelley & Heatherton, 2011*; *Herman, Polivy & Esses, 1987*; *Mills & Palandra, 2008*) followed by ad libitum, disinhibited eating of ice-cream. After drinking an entire 15-ounce milkshake, participants were instructed to sample three flavors of ice cream under the guise of a taste test. Previous studies have demonstrated that dieters whose diets are broken tend to eat more ice cream, but there are individual differences in amount

of ice cream consumed (*Demos, Kelley & Heatherton, 2011*). The total amount (grams) of ice cream consumed was measured using an Ozeri Pro Digital Kitchen Food Scale (Ozeri USA, San Diego, CA). Previous research using this laboratory-based manipulation has consistently demonstrated that following a milkshake preload, dieters will proceed to eat significantly more ice cream than those participants whose diets are not broken (e.g., *Demos, Kelley & Heatherton, 2011*; *Heatherton & Baumeister, 1991*; *Herman & Mack, 1975*; *Tomiyama et al., 2009*).

## Analysis of fMRI data

All analyses of neuroimaging data were conducted using SPM8 (Wellcome Department of Cognitive Neurology, London, England), along with tools for batch preprocessing and analysis (available at http://github.com/ddwagner/SPM8w). First, we carried out standard preprocessing procedures for both inhibitory control and cue reactivity tasks, to account for motion related artifact (by including six motion regressors in first-level GLM model specifications) and increase images' signal-to-noise ratio via spatial smoothing (8-mm FWHM kernel). Between the two tasks, five participants' data were excluded due to excessive motion, defined as $\geq 2$ instances of $\geq 2$ millimeters of movement in the $x, y$, or $z$ plane (resulting $N = 27$ for subsequent analysis). In the case of the main analysis (relating DLPFC activity to subsequent food cue reactivity), there were 26 participants who had complete data across those two measures. So, for those reported analyses, $N = 26$.

Brain data for the two tasks were separately modeled and analyzed. First, for the effortful self-control task, we ran a whole-brain univariate GLM analysis at the first (subject) level, with two regressors specifying when words appeared on the screen and when words were absent (i.e., only the film was playing). The main contrast of interest (words present vs. absent) identified period of the task when participants were actively engaging in self-control. Group-level T-maps were then generated using random effects analysis of all subjects' word present-versus-word absent contrast images. To correct for multiple comparisons, we performed cluster-based correction via the AFNI tool *3dClustSim*, which performed 10,000 simulations using the mean smoothness estimated from residual images (obtained from each participant's first-level GLM) and using spatial autocorrelation function parameters (via *3dClustSim*'s "acf" option)—as per recent recommendations (*Cox et al., 2017*). These simulations returned a minimum cluster size of 25 voxels at an uncorrected $p < .001$ threshold required for a cluster-level false positive discovery rate of $p < .05$ (thresholded map available at: https://neurovault.org/collections/EDTWSFKM/images/60660/).

We first computed the cluster thresholded map from the words-versus-film contrast during the effortful self-control task—indexing brain activity while participants were resisting the urge to read words whenever they appeared on the screen. This map revealed activity in regions of both prefrontal and parietal cortex associated with attention and self-control (see Fig. 1 for thresholded T-map, indicated by orange/yellow gradient; Table 1 for a full list of supra-threshold regions of peak activity). To test our hypothesis that lateral prefrontal cortex would be engaged during effortful self-control, we extracted parameter estimates from the cluster in the dorsolateral prefrontal cortex (peak MNI coordinates

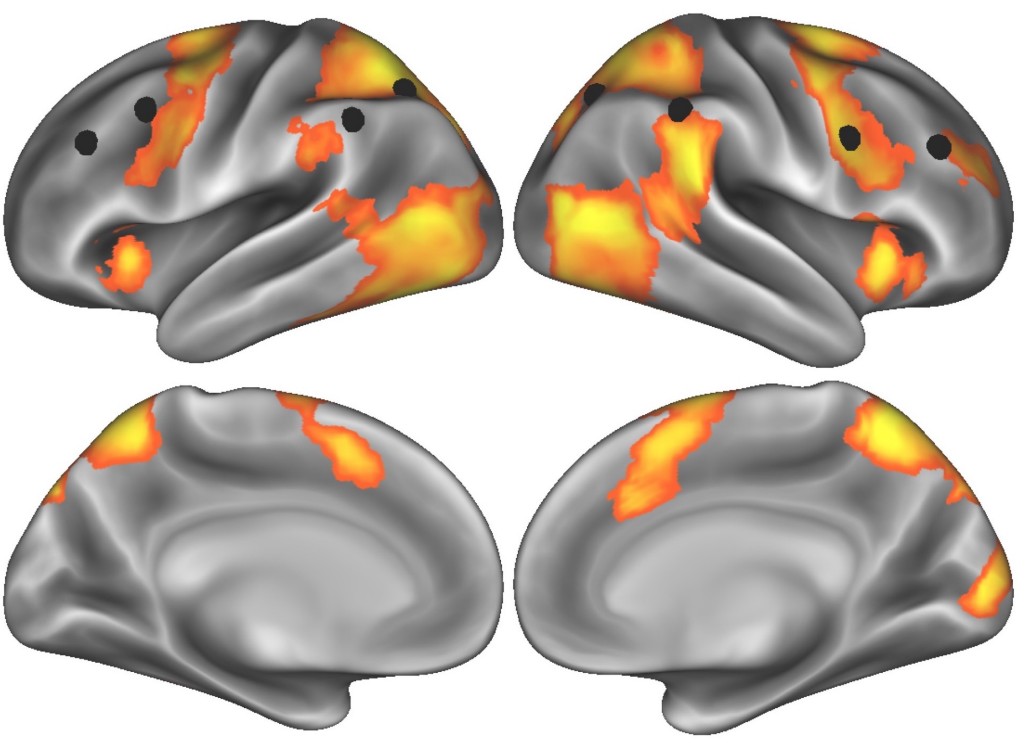

**Figure 1** Task-elicited brain activity related to effortful self-control, with eight nodes comprising the frontoparietal control network (*Power et al., 2011*) indicated by black spheres.

**Table 1** Supra-threshold brain regions activated during effortful self-control from the first fMRI task.

| Region[a] | MNI coordinates of peak voxel | # voxels | Effect size (*t* value) |
|---|---|---|---|
| Right dorsolateral prefrontal cortex | 30, 42, 24 | 206 | 5.42 |
| Right superior frontal gyrus | 21, −3, 63 | 2,207 | 8.17 |
| Right anterior insula/frontal operculum | 36, 18, 3 | 329 | 7.09 |
| Left superior parietal lobe | −24, −54, 57 | 6,788 | 13.14 |
| Right cerebellum | 9, −78, −24 | 192 | 7.29 |
| Left anterior insula/frontal operculum | −30, 21, 3 | 111 | 7.03 |
| Right occipital pole | 9, −90, 9 | 229 | 7.53 |

**Notes.**
[a] Unless otherwise indicated, labels are taken from the Harvard–Oxford cortical atlas (*Desikan et al., 2006*).

$X = 30$, $Y = 42$, $Z = 24$) using a spherical ROI with a 6-millimeter radius. Additionally, given the role the frontoparietal network plays in cognitive control generally, and self-regulation of eating more specifically (*Lopez et al., 2017*), we extracted standardized, aggregate activity across eight nodes of the frontoparietal network (*Power et al., 2011*) and

used this system-level activity as an *a priori* brain measure to relate to subsequent food cue activity.

Next, to estimate activity elicited by food cues when participants were watching commercials, we ran a separate univariate GLM analysis modeling food events (i.e., whenever featured food items appeared on the screen during a food commercial) and control (product) events. Next, contrast images were calculated to compare activity when food events occurred relative to that when control events occurred. Other commercial content (i.e., all time points when products were not featured/visible on the screen) were modeled as baseline. We extracted food cue related activity from these contrast images using independently defined, *a priori* regions/systems of interest, namely: eight nodes of the frontoparietal control network, which supports flexible exertion of self-control (*Dosenbach et al., 2007*; *Power et al., 2011*); and two regions in the reward system, OFC and bilateral ventral striatum, which reliably activate to appetizing food images (MNI coordinates for OFC: $-30, 33, -18$ ; VS: $\pm9, 3, -6$ (taken from *Wagner et al., 2013*). We selected eight nodes in the frontoparietal control network to remain consistent with our prior work showing that aggregate activity across these eight nodes, versus activity in a larger set of frontoparietal nodes, was most predictive of dieters' self-control success and failure in daily life (*Lopez et al., 2017*). However, in order to investigate whether this effect generalizes across a larger set of nodes we used 31 frontoparietal nodes from a recently published parcellation study within our group based on a large, independent sample ($N = 828$; *Huckins et al., 2019*) to re-compute regulation–reward balance scores. The findings presented in the main text largely replicated (for more details see Supplementary Materials for all results from this analysis). Food-cue specific activity in both control and reward regions was extracted using a spherical ROI with a 6-millimeter radius.

For our main analysis, we related task-elicited activity in DLPFC and mean recruitment of the frontoparietal network from the effortful self-control task to the relative recruitment of frontoparietal control network (vs. reward system) during food cue exposure in the second fMRI task. This relative recruitment was captured by regulation–reward balance scores, which were calculated on a subject by subject basis by taking the difference of standardized (i.e., *Z*-scored), averaged activity in the frontoparietal control network and activity in reward regions (i.e., OFC and bilateral ventral striatum). Higher balance scores represent relatively greater recruitment of the frontoparietal control network, whereas lower balance scores represent relatively greater recruitment of reward regions; this procedure and operationalization of a brain-based balance measure followed that of Lopez and colleagues' (*2017*) study.

The main data file used to run all models and compute all statistics is available at the following repository on the *Open Science Framework*: https://osf.io/qpt3f/.

## RESULTS

First, as a manipulation check, we examined participants' ratings of their experience of the first (effortful self-control) task they completed in the scanning session. On average, participants found the task to be difficult, as inferred from a one-sample *t*-test against the scale's midpoint value of 4, $M = 4.93$ (Difference from test value: 0.93, 95% CI for

difference: 0.45, 1.40), $SD = 1.21$, $t(26) = 3.99$, $p < .001$, Cohen's $d = 0.77$; this value was significantly greater than dieters in a control condition from a similar study, $t(26) = 3.09$, Cohen's $d = 0.60$, $p = .005$. The self-reported measures of difficulty and tiredness were not correlated with one another, $r(25) = 0.004$, $t = 0.02$, $p = .985$, so they were entered in as separate covariates in multiple regressions models described below. Next, we observed that across all 8 nodes of the FP network, there was significant activity in the first fMRI task during effortful self-control (i.e., words versus film contrast), mean parameter estimate = 0.159 (95% CI [0.092–0.226]), SD = 0.169, $t(26) = 4.89$, Cohen's $d = 0.941$, $p < .001$.

To relate brain activity during effortful self-control to subsequent food cue reactivity, we first calculated the correlation between brain activity in this region and the relative balance of activity in frontoparietal (vs. reward) regions during exposure to food commercials. There was a significant negative association, such that those participants who more readily recruited the DLPFC during the effortful self-control task had lower balance scores when viewing food commercials, $r(24) = -0.574$ (95% bootstrapped CI with 10,000 iterations: $-0.753, -0.310$), $b = -2.30$, $p = .002$ (see Fig. 2). Importantly, this relationship held when controlling for participants' self-reported difficulty and tiredness when they performed the effortful self-control task, as well as participants' body mass index (BMI), $b = -2.538$ (95% CI: $-4.08, -1.00$), $t(19) = -3.445$, $p = .003$. In this multiple regression model, neither task difficulty ($b = -0.021$, $t(19) = -0.261$, $p = .797$) nor tiredness ($b = 0.042$, $t(19) = 0.630$, $p = .536$) was associated with balance scores. Additionally, upon examining zero-order correlations, self-reported difficulty was not associated with DLPFC activity during the effortful self-control task ($r = 0.062$, $p = .760$) or balance scores during food cue reactivity ($r = -0.103$, $p = .615$). This was also true for self-reported tiredness, which did not correlate with DLPFC activity ($r = 0.206$, $p = .302$) or balance scores ($r = -0.003$, $p = .988$).

Next, consistent with the negative association between DLPFC activity during effortful self-control and balance scores during subsequent exposure to food cues, there was also a significant negative relationship between average recruitment of the frontoparietal network activation during effortful self-control and balance scores in the cue reactivity task, $r(24) = -0.403$ (95% bootstrapped CI with 10,000 iterations: $-0.741, -0.032$), $b = -0.302$, $p = .041$, and this relationship remained after controlling for self-reported difficulty and tiredness and BMI, $b = -0.339$ (95% CI [$-0.67, -0.01$]), $t(19) = -2.168$, $p = .043$. Critically, the association between frontoparietal network recruitment during effortful self-control was only observed with balance scores during food cue-exposure. Specifically, there was no relationship between FP network activity during self-control and subsequent FP network activity during cue-reactivity, $r(24) = -0.206$, $t = -1.03$, $p = .313$, nor was initial FP network recruitment associated with reward system activity during cue reactivity, $r(24) = -0.081$, $t = 0.40$, $p = .693$. To rule out influences of nuisance variables (e.g., differences in global BOLD signal between individuals) we extracted parameter estimates based on an anatomical mask of the hippocampus (a region not engaged by either task) and found no relationship during the first self-control task and balance scores in the cue reactivity task, $r(24) = -0.09$, $t = -0.444$, $p = .661$.

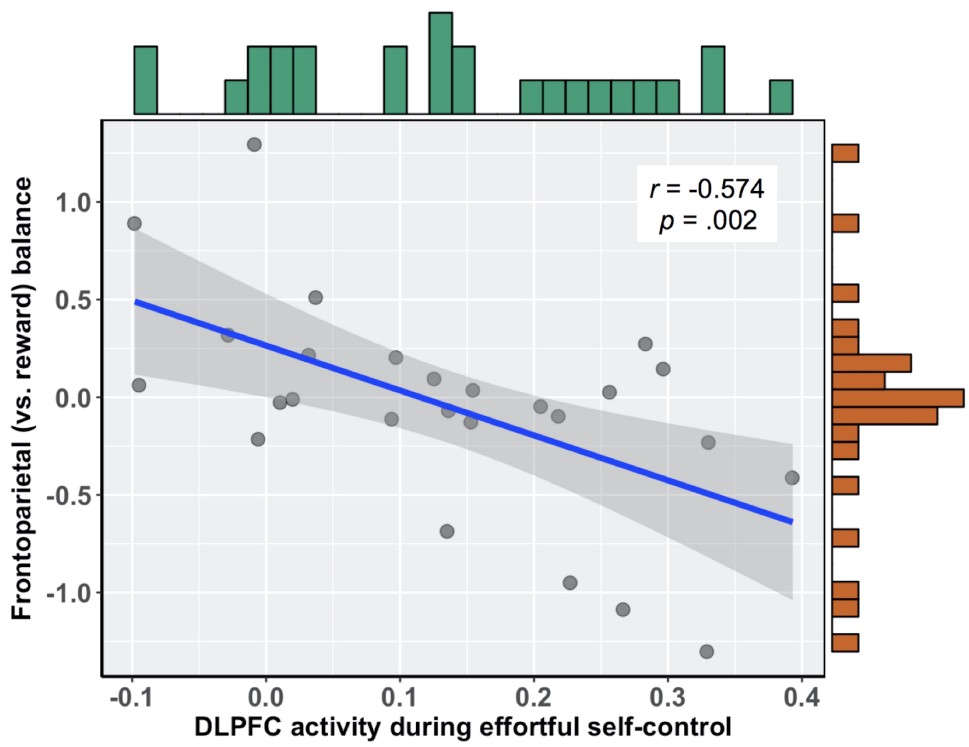

**Figure 2** Negative association between DLPFC activity during effortful self-control and subsequent frontoparietal (vs. reward) balance scores during exposure to food commercials.

Finally, we examined links between activity in the effortful self-control/cue reactivity tasks and ad libitum ice-cream eating from the behavioral session. On average, participants consumed 71.1 grams of ice-cream ($SD = 40.8$, range: 9–158 grams), and there was a positive (albeit non-significant) correlation between recruitment of DLPFC during effortful self-control and grams of ice-cream consumed, $r(24) = 0.319$, $t = 1.647$, $p = .112$. There was also a negative, non-significant relationship between balance scores and grams of ice-cream consumed, $r(24) = -0.309$, $t = -1.594$, $p = .124$.

## DISCUSSION

Results from this study indicated that individual differences in chronic dieters' recruitment of brain regions associated with cognitive control (i.e., DLPFC and frontoparietal network) while performing an effortful self-control task was subsequently associated with the balance of activity between brain regions implicated in self-control and reward during exposure to food commercials. Consistent with our hypothesis, greater recruitment of DLPFC—as well as the frontoparietal network as a whole—during effortful self-control was associated with less recruitment of the frontoparietal network and more reward related activity whenever food items appeared on the screen in the cue reactivity task. This suggests that more (or less) brain activity during an initial task requiring self-control exertion may serve to characterize the different response profiles that dieters may show when exposed to appetitive food cues.

Notably, we only observed a significant, negative relationship between FP recruitment during effortful self-control and subsequent balance scores during exposure to food commercials. There was no such relationship between control-related FP recruitment in the first task and activity in either FP or reward systems, respectively, during food cue reactivity (second task). This is consistent with theorizing that suggests taking both impulsive and inhibitory processes into account to characterize self-control outcomes (*James, 1890*; *Kotabe & Hofmann, 2015*; *Lewin, 1951*), as well as recent evidence showing that the relative balance of control (versus reward) activity was predictive of self-regulation outcomes—whereas activity in either system alone was not (*Lopez et al., 2017*).

The present findings also substantiate and extend previous studies that have shown that when participants are engaged in effortful self-control tasks they subsequently show *reduced* brain activity in lateral prefrontal cortex when performing cognitive tasks (*Friese et al., 2013*; e.g., *Luethi et al., 2016*). For example, in a study by Friese and colleagues (2013), those participants who were randomly assigned to suppress their emotions when viewing highly distressing, negatively-valenced stimuli showed reduced recruitment of lateral prefrontal cortex during a subsequent Stroop task, compared to those participants in the control condition (*Friese et al., 2013*). However, one of the features that differentiates our study from previous work is that we focused on dieters' neural responses to appetitive food cues, following previous self-control exertion. This approach has the advantage of looking at a motivationally relevant class of stimuli in a population that, by definition, is chronically engaged in self-regulation and inherently motivated to regulate their responses. However, we would be remiss if we did not acknowledge that it remains an open question as to whether the effect of prior self-control exertion on neural responses to food cues reflects either: (1) a reduction in self-control capacity, as a resource-based account would predict (e.g., *Baumeister, Muraven & Tice, 2000*); or (2) the fact the those participants who experience the task as more effortful are simply less motivated to continue regulating in a secondary task, as per motivation-based accounts (e.g., *Inzlicht & Schmeichel, 2012*).Without additional measures, the present study cannot disentangle these two views (see above, with associated references, for the most recent theorizing and findings related to potential mechanisms underlying self-control fatigue and lapses).

In addition, we examined the relationship between these neural measures and ad libitum eating of ice cream following a diet breaking episode in a separate experiment. Although the relationships were in the expected direction (i.e., greater grams of ice cream eaten among participants who, on average, showed more FP recruitment during effortful self-control and also lower balance scores), the overall correlations with neural measures were not significant. Whether this reflects a true null effect, or instead that the mechanisms underlying individual differences in neural activity we observed following self-control exertion do not translate to disinhibited eating following a diet breaking episode, remains an open question.

There are several strengths to our study design and approach. First, our overall analysis approach—linking brain activity from an effortful self-control task to activity in a subsequent task that *also* calls for spontaneous regulation of prepotent responses—is arguably more ecologically valid than previous studies that have examined effortful

self-control. Although speculative, we would argue that the sequence and nature of the two self-control tasks used here (i.e., an intensive, initial exertion of effort to inhibit, followed by unpredictable (but no less needed) instances that also require self-control) may mimic dieters' experiences in daily life. For example, a dieter might exert self-control to block out distraction during an intense time of study or work. And, the extent to which they find this sustained inhibition taxing, they may have little self-control capacity to call upon when they are suddenly faced with a dessert tray at a restaurant later that day.

Second, the current study's design may reliably identify those individuals who experience more (or less) success adhering to their dieting goals. For example, a dieter who more readily recruits certain cognitive control regions (i.e., DLPFC) during an effortful self-control task may be actively maintaining the task set (i.e., "avoid reading the words"), but such task engagement may be more cognitively demanding and render the frontoparietal network less able to exert control over the reward system during future exposure to food cues (cf. *Heatherton & Wagner, 2011*). Indeed, the effects observed here in the appetitive domain are consistent with other studies that have showed reductions in activity in prefrontal cortex—and accompanying task deficits in various cognitive tasks—following self-control exertion (*Friese et al., 2013*; *Luethi et al., 2016*; e.g., *Persson, Larsson & Reuter-Lorenz, 2013*). The present findings are also consistent with predictions made by models emphasizing the role of self-regulatory fatigue in affecting self-control outcomes (*Evans, Boggero & Segerstrom, 2015*; *Hofmann et al., 2012*; *Wilkowski et al., 2018*), and those that highlight the limitations of cognitive control processes more broadly (*Shenhav et al., 2017*).

Additionally, if it is true that greater recruitment of brain regions during an effortful self-control task can undermine future self-control attempts, then it means that more control-related activity (during initial exertion) is not necessarily conducive to dieters' self-regulatory goals. Indeed, and consistent with prior behavioral work on limited self-regulatory capacity (e.g., *Baumeister et al., 1998*), control-related activity during self-control exertion may reflect task difficulty or effort, potentially leading to fatigue and dieters' inability to later recruit self-control regions when faced with (subsequent) appetitive temptations. But, since participants' self-reported difficulty/fatigue ratings did not correlate with brain activity during effortful self-control, or with the frontoparietal (vs. reward) balance measure, it is possible that dieters do not have conscious awareness or insight into how much they are affected by exertions of self-control (or not). Future work may consider improving dieters' self-awareness about their susceptibility, neural or otherwise, to improve adherence to self-regulatory goals over time. Indeed, others have made similar arguments about targeting people's self-awareness and insight in the context treatments for drug addiction (*Goldstein et al., 2009*).

Despite the implications discussed thus far, the current study has some limitations that are worth mentioning. First, the self-control task we administered was somewhat short in length (i.e., 7 min), so it is not clear how long effects of effortful self-control on cue reactivity might persist. Second, even though our participants gave ratings of difficulty and fatigue during the effortful self-control task, we did not have an independent measure of participants' objective performance on the task (i.e., how successful they were in avoiding reading the words on the screen). A future study would benefit from incorporating

eye-tracking or another validation measure to calculate, on a subject-by-subject basis, successful inhibition of the impulse to read the words. And even though we used a validated diet-breaking procedure for the behavioral session, using a self-control exertion task would have led to more construct validity and may help explain why we did not find a relationship between the brain data and overeating after the milkshake preload.

We should also note that, in comparison to behavioral work, the sample size was relatively small and thus was not powered for the detection of moderate correlations. This may place a limit on our ability to detect small to moderate effects of self-control exertion (e.g., we did not observe any relationship between brain activity and ice-cream eating). Although to remain consistent with prior work, with recruitment necessarily constrained to female dieters, future work would benefit from either expanding beyond this population in order to increase sample sizes and test generalizability to other populations.

Another possibility is that our sample, on average, showed relatively little ice cream consumption (mean = 71.1 grams), as well as a more restricted range ($SD = 40.8$ grams) compared to previous studies that also measured disinhibited ice-cream eating in dieters (cf. mean values of 71.7–211.2 grams and $SD$ values of 57.4–123.8 in (*Vohs & Heatherton, 2000*). Also, in order to establish equivalent levels of motivations to regulate eating, we only recruited from the dieting population. So, the boundary conditions and generalizability of this effect need to be tested in other populations, as it is probable that the nature of self-regulatory goals and other related factors may modulate patterns of brain activity during effortful self-control (and subsequent food cue reactivity). Lastly, one overall caveat to the present work is that all reported findings are correlational, so no strong claims can be made as far as the directionality of the observed effects.

## CONCLUSIONS

To conclude, we probed the neural mechanisms of repeated self-control exertion among dieters, using a reasonably naturalistic dual-task paradigm that coupled an initial, effortful self-control task with subsequent exposure to appetitive food cues as presented in real-world food commercials. We extended past work, which took a brain-as-predictor approach (*Berkman & Falk, 2013*) using the balance between prefrontal and reward related responses during food cue exposure to predict dietary failure (*Lopez et al., 2014*), by using this same measure to examine how prior exertion of self-control may lead to impaired dietary self-control during exposure to tempting food cues. In doing so, we tried to adopt a health neuroscience framework, a primary aim of which is to better characterize health risk behaviors—including patterns of over-eating that lead to obesity—in order to alleviate the various burdens, medical and otherwise, associated with preventable chronic disease (*Hall et al., 2018*). Future intervention studies and clinical trials may benefit from taking a similar approach and target such brain-based risk factor, with the goal of modulating brain responses during effortful self-control and cue reactivity, and possibly producing changes in real world eating behaviors.

### Funding

This work was supported by the National Institutes of Health (No. R01DA022582). The funders had no role in study design, data collection and analysis, decision to publish, or preparation of the manuscript.

### Grant Disclosures

The following grant information was disclosed by the authors:
National Institutes of Health: No. R01DA022582.

### Competing Interests

The authors declare there are no competing interests.

### Author Contributions

- Richard B. Lopez conceived and designed the experiments, performed the experiments, analyzed the data, contributed reagents/materials/analysis tools, prepared figures and/or tables, authored or reviewed drafts of the paper, approved the final draft.
- Andrea L. Courtney conceived and designed the experiments, performed the experiments, analyzed the data, contributed reagents/materials/analysis tools, authored or reviewed drafts of the paper, approved the final draft.
- Dylan D. Wagner analyzed the data, contributed reagents/materials/analysis tools, authored or reviewed drafts of the paper, approved the final draft, developed and maintained the software (spm8w) used for neuroimaging analyses.

### Human Ethics

The following information was supplied relating to ethical approvals (i.e., approving body and any reference numbers):

All participants gave informed consent in accordance with guidelines set by the Committee for the Protection of Human Subjects at Dartmouth College, and were fully debriefed at the end of the study (IRB approval #20325).

### Data Availability

Raw data are available at the Open Science Framework: Lopez, R.B., Courtney, A.L., & Wagner, D.D. (2019). Neural correlates of effortful self-control in dieters. Retrieved from https://osf.io/qpt3f/.

### Supplemental Information

Supplemental information for this article can be found online at http://dx.doi.org/10.7717/peerj.6550#supplemental-information.

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
