# Peer review of "Recruitment of cognitive control regions during effortful self-control is associated with altered brain activity in control and reward systems in dieters during subsequent exposure to food commercials"

_PeerJ, doi:10.7717/peerj.6550_

## Round 0.1 · original submission · Major Revisions

Your manuscript has been revised by three independent reviewers. Two of them recommended minor revision, but reviewer 2 has indicated a major revision is required. To facilitate the re-review process, please, indicate in the rebuttal letter where the changes were introduced to attend the criticism of the reviewers.

Reviewer 1 ·

Basic reporting

See my comment below.

Experimental design

See my comment below.

Validity of the findings

See my comment below.

Additional comments

In this study, the authors examined the neural mechanisms through which the exertion of self-control magnifies subsequent self-control failures. Specifically, the authors tested whether lateral PFC activation during self-control exertion would predict a reduced balance between this PFC region and neural ‘reward’ regions, and would predict increased self-control failure in the form of ice-cream consumption among a sample of female dieters. The authors found evidence that DLPFC recruitment during self-control exertion was subsequently associated with less connectivity between the reward circuit and the lateral PFC.
I’ve been waiting for a study like this (i.e., one that uses fMRI during a reward-based self-regulatory challenge *and* afterwards during an appetitive task) to be run for years, so glad to finally see one! There’s a lot to like here: combining neural predictors with real-world behaviors, examining neural correlates of self-control exertion, recruiting a sample of people in an ecologically-valid chronic self-control challenge (i.e., dieting), and a theory-derived set of predictions on a critically-important and impactful subject. The writing is impeccable and the logical flow of the paper is fluid.
Below, I list some comments and questions for the authors:
• The second citation in your Intro includes the first author’s first name and middle initial. This also happens in some other parts of the manuscript.
• In your hypotheses section of your Intro, you should mention the context in which you will be measuring lateral PFC recruitment.
• At the canonical 80% power level, your sample size of 27 is too small to detect between-subjects correlations smaller than r = .52. We rarely see such large effect sizes in human behavior. As such, the analyses in this paper are almost certainly severely underpowered and this needs to be more clearly stated in the Methods, Discussion, and the N = 27 sample size must be mentioned in the Abstract to engender the appropriate level of skepticism in your readers. As it stands, your reduced sample size is squirreled away in the ‘Analysis of fMRI Data’ sub-section.
• Unless the authors believe that all participants were 100% successful in not reading the distractor words during the self-control challenge task, their manipulation of self-control engagement is confounded with lexical processing.
• How were the DLPFC ROI peak coordinates determined?
• Please depict the 8 nodes of the frontoparietal network and the ROIs you built around them. Maybe as a supplemental figure.
• Were the OFC/striatum ROIs spherical or anatomical?
• Why is one of your p-values labeled as ‘two-tailed p = .005’? Are the other p-values one-tailed?
• How were the “balance scores” calculated?
• Some degrees of freedom seem off. You should have 25 degrees of freedom for your bivariate correlations, but I see “r(24) = -0.574” and then when 3 covariates were included, which should’ve reduced your dfs to 22, I see “t(19) = -3.445”. Am I missing something?
• Please report exact association statistics and p-values, not ranges (e.g., “p’s ≥ 0.536”).
• Why are 95% CIs reported for some findings but not others?
• “participants 342 consumed 71.1 grams of ice-cream (SD = 40.8, range: 9–158 grams)” Wow!
• Were the food commercials equivalent to the non-food commercials in important aspects such as arousal or valence? If not, this may not be a story about food per se, but about arousing/rewarding stimuli more generally.
• You refer to p-values of .112 and .124 as “trending”. C’mon, that’s way too much of a stretch. Please recall that trending can mean trending *away* from significance just as much as it can mean *towards* significance. I think these just need to be called the null results that they are. I’m bummed that these weren’t significant (and I bet they would be if you had a larger sample). What is the distribution of the ice-cream scores like? If it’s positively skewed, does a log-transformation help?
• There’s no way you have the power to accurately test this, but I’m curious of the association between DLPFC recruitment during self-control and ice cream consumption is mediated by balance scores during food commercial viewing.
• Correlational nature of the study is not mentioned.

Reviewer 2 ·

Basic reporting

This article is well written and contained sufficient background and details.

My main critique of the reporting has to do with the focus on the ego depletion debate in the introduction. The authors acknowledge in the discussion section that “the present study cannot disentangle these two views” (p. 16). While I think it is important to acknowledge this debate, I think the introduction should focus more on the specific questions and gaps in knowledge that this paper can address. I would have been more convinced if the authors spent more time explaining why fatigue/depletion (or whatever you want to call it) of cognitive resources is important to study independent of this debate. For example, spending more time discussing evidence that self-control failure follows earlier attempts at effortful self-control, particularly in the domain of dieting, would be more convincing.

Experimental design

The experimental design is well thought out. Both tasks are more ecologically valid than typical fMRI tasks. My primary concerns are with the small sample size, relatively short initial self-control tasks, and lack of a self-control task in the one week follow-up. If the authors were interested in how initial self-control attempts influence subsequent eating behavior, wouldn't it make sense to include a self-control task before the milkshake and ice cream taste tests? The authors do a good job acknowledging the first two issues and I'm not sure if anything can be done at this point to address the third point.

Finally, I thought the description of the frontoparietal network nodes was lacking some crucial details. Unless I’m mistaken, I believe there are 25 frontoparietal nodes in the Power et al. (2011) parcellation. If the authors only used a subset of these nodes they should explain how they were chosen. It would also be important to note if the results hold when including the full frontoparietal system.

Validity of the findings

The current study compared activation across two tasks and applied a network approach to investigate how activation across multiple brain regions might relate to how dieters respond to food cues. A key strength of this approach is using both task-defined ROIs and a priori system labels. This research takes a novel approach and provides some interesting evidence that recruitment of neural self-control resources may be reduced over time due to depletion/fatigue/reduced motivation. I have two primary concerns with the interpretation of results:

First, I had a hard time following the cognitive efficiency hypothesis as it relates to this data. If weaker activation is actually indicative of better processing/self-control, then why do people who have greater activation in the first task have weaker activation in the second task? Are people getting better at self-control over time? This seems counterintuitive and only makes the results harder to make sense of.

Second, do the results hold if the authors recompute the balance scores excluding ventral striatum? I ask this primarily because striatum is not just a reward region—it is also implicated in response inhibition. It seems problematic to argue that striatum is a reward region when it is also involved in response inhibition.

Reviewer 3 ·

Basic reporting

This manuscript was easy to follow and clear in its writing. The Introduction covered the relevant literature adequately, although I would have liked to see a more explicit statement on how the current work differs from Wagner et al. (2013). (I was able to piece it together by re-reading, but a more explicit statement of the differences would have saved this reader some time and effort.) The figures were relevant and clear, and the data was made available for review.

Experimental design

This was original primary research, and the research question was well defined, relevant, and meaningful. As mentioned above, a more explicit statement of how this research differs from past research, and how it fills an identified knowledge gap, was lacking.

Methods were described in sufficient detail and the work appears to have been performed to high ethical and technical standards.

Validity of the findings

The data analysis was sound and interesting. I think it is inappropriate, however, to refer to a “positive trending correlation” (lines 342 and 343). The same goes for the “trending” correlation reported in lines 344 to 345. Simpler to say the correlations were non-significant but in the predicted direction (as the authors go on to do in line 389). We don’t know which way they were “trending.”

Further, I thought the Discussion section included a balanced assessment of the strengths and weaknesses of this work.

---

## Round 0.2 · accepted · Accept

I have read your rebuttal letter and your tracked revised manuscript with details. I realize that you have addressed all the points specified by the reviewers.

#